# Effect of New Complete Dentures and Simple Dietary Advice on Cognitive Screening Test among Edentulous Older Adults: A Randomized Controlled Trial

**DOI:** 10.3390/jcm12144709

**Published:** 2023-07-16

**Authors:** Yuriko Komagamine, Hiroyuki Suzuki, Maiko Iwaki, Shunsuke Minakuchi, Manabu Kanazawa

**Affiliations:** 1Gerodontology and Oral Rehabilitation, Graduate School of Medical and Dental Sciences, Tokyo Medical and Dental University, Tokyo 113-8549, Japan; y.komagamine.gerd@tmd.ac.jp (Y.K.); h.suzuki.gerd@tmd.ac.jp (H.S.); s.minakuchi.gerd@tmd.ac.jp (S.M.); 2Didital Dentistry, Graduate School of Medical and Dental Sciences, Tokyo Medical and Dental University, Tokyo 113-8549, Japan; m.iwaki.gerd@tmd.ac.jp

**Keywords:** complete dentures, cognitive function, mild cognitive impairment, dietary intervention, nutrition, edentulousness

## Abstract

Mild cognitive impairment (MCI), including memory loss, has been attracting attention in Japan. This study assessed the effect of new complete dentures provision alone and with dietary intervention on cognitive functions assessed using the Japanese version of the Montreal Cognitive Assessment (MoCA-J). A randomized controlled trial was conducted with 70 older adults who required new complete dentures. The participants had new complete dentures fabricated and were randomly classified into the intervention or control group. The intervention group received simple dietary advice, and the control group only received denture care advice. Cognitive function was assessed using the MoCA-J before and at 3 and 6 months after treatment. The between-group comparison and within-group comparison were analyzed. No significant differences were reported for comparisons between the intervention and control groups. A significant increase was revealed in the within-group comparisons for the total scores between the 3- and 6-month assessments (*p* = 0.002) and between the baseline and 6-month assessments (*p* = 0.012) in the intervention group. In the control group, a significant increase in the total scores was not revealed between any of evaluation period. Complete denture replacement combined with simple dietary intervention may help improve MoCA-J scores in edentulous older adults.

## 1. Introduction

In Japan, dementia is the leading cause for older adults requiring nursing care. Recently, mild cognitive impairment (MCI), including memory loss, which is a precursor symptom of dementia, has been attracting attention [1]. MCI is likely to progress to dementia if left untreated; therefore, early detection and treatment at the occurrence of MCI are desirable.

In a 4-year longitudinal study that investigated the association between denture wear and MCI onset as assessed by the Mini-Mental State Examination (MMSE) [2], Yamamoto et al. [3] reported MCI onset risk 4 years after assessment at baseline was 2.70 times higher in the denture-wearing group of edentulous adults. Furthermore, developing dementia risk 4 years after assessment at baseline was 4.75 times higher in the non-denture-wearing group, indicating denture wear may improve masticatory function and prevent cognitive impairment. Morokuma et al. [4] reported that brain function activation was observed in edentulous adults before and after adjustments of their complete dentures using an electroencephalography measurement device. Several previous studies observed brain activity during clenching in edentulous adults wearing old and new dentures, respectively, and reported that brain activity in the primary sensorimotor cortex was increased during post-denture adjustments or wearing new dentures [5,6,7]. These studies revealed that wearing well-fitted complete dentures may prevent cognitive decline in edentulous adults as assessed by the MMSE. However, to date, no study has evaluated the effect of wearing new complete dentures on Montreal Cognitive Assessment (MoCA) scores [8] of edentulous older adults.

Seraj et al. [9] reported an association between a low intake of omega-3 fatty acids, antioxidants, and vitamin B12 and cognitive impairment, noting that vitamin B especially influences dementia and cognitive impairment. Several previous studies have shown that patients who need dentures do not change their eating habits and therefore do not change their dietary and nutritional intake if only new dentures are provided without dietary intervention [10,11,12]. Instead, the provision of new dentures combined with dietary intervention improved dietary and nutritional intake in edentulous patients.

We previously assessed if the combination of simple dietary intervention provided by dentists would improve nutritional and dietary intake in edentulous patients after new complete dentures [13,14]. In the study, regarding dietary intake, at the 3-month assessment, the intervention group showed a significantly greater intake of chicken, fish with bones, carrots, and pumpkins compared to the control group. Regarding nutritional intake, at the 3-month assessment, the intake of several nutrients such as proteins, minerals, and vitamins was significantly higher in the intervention group than in the control group. Furthermore, at the 6-month assessment, animal protein and vitamin B12 intakes were significantly higher in the control group although the effect of simple dietary advice decreased. Therefore, dietary and nutritional intake including vitamin B12 improved in edentulous patients after new complete dentures when combined with simple dietary intervention provided by dentists [13,14], which might have a greater effect on cognitive function than complete denture replacement alone. However, there had been no study that assessed the effect of dietary advice combined with complete denture replacement on cognitive function.

Therefore, in this study, we assessed the effects of new complete denture provision alone and combined with dietary intervention on the scores of the cognitive screening test, the Japanese version of the MoCA (MoCA-J) [15]. This study’s null hypothesis was that in edentulous patients there would be no difference in MoCA-J score improvement for those who received dietary intervention combined with complete denture replacement and those with only complete denture replacement.

## 2. Materials and Methods

### 2.1. Trial Design

This study was a double-blinded, randomized controlled, clinical trial, which complied with the requirements of the 2010 Consolidated Standards for Reporting Trials (CONSORT) Statement. The primary outcome results have already been reported in another study [13], and this study reports the results of one of the secondary outcomes. The detailed information of this study’s protocol is described in a previous publications [13,14]. The trial protocol was approved by the Ethics Committee of the Faculty of Dentistry, Tokyo Medical and Dental University (TMDU, Resister number 1144), and registered in the University Hospital Medical Information Network (UMIN) Center (UMIN-CTR Clinical Trial, Unique Trial Number: UMIN000017879). All the study participants provided written informed consent.

### 2.2. Participants

#### 2.2.1. Inclusion Criteria

Each patient fulfilled the following requirements: edentulous jaw status; wanted to replace their dentures due to dental problems; able to visit the hospital on their own; and able to understand written and spoken Japanese and respond to a questionnaire.

#### 2.2.2. Exclusion Criteria

Patients were excluded if they fulfilled at least one of the following: had an infectious disease; received professional nutritional counseling; lived in an institution where they could not control their diet; had an oral disorder (xerostomia, temporomandibular joints arthrosis, or oral dyskinesia) or psychiatric disorder or dementia diagnosis; or suffered from severe cardiovascular, metabolic, or other diseases that required specialist dietary restrictions. There were no age restrictions.

### 2.3. Intervention

All participants were randomly assigned to the dietary intervention or control group. Clinicians fabricated conventional complete dentures for the upper and lower jaws.

The dietary intervention group received new complete dentures and 20 min of brief dietary instruction. The brief dietary instruction was orally performed by the dentist using a uniform pamphlet. If the participants did not cook, the person who cooked for the participants received instruction by telephone. The pamphlet was a geriatric version of the Japanese food guide published by the Japanese Ministry of Agriculture, Forestry and Fisheries that indicates ‘what’ and ‘how much’ should be eaten in a day. It uses a graphic of a spinning top, to illustrate the proportions of each food group that should be consumed daily, similar to the ‘food pyramid’ typically presented to Western audiences [16,17].

On the other hand, the control group received new complete dentures and 20 min of instruction on how to care for their dentures from a pamphlet created based on the guideline of the American College of Prosthodontics [18]. Both of the instructions were given twice: on the day of the trial fitting and the day of complete denture delivery.

### 2.4. Outcome

Cognitive function was assessed using the MoCA-J, a 30-point test, with higher scores indicating better cognitive function [15]. A patient was diagnosed with MCI if the score was less than 26 [15]. One point was added if a participant had less than 12 years of education. The outcomes were assessed at baseline and 3 and 6 months after the final denture adjustment.

The MoCA-J consists of eight items including 12 tasks. The eight items are “Visuospatial /Executive”, “Naming”, “Memory”, “Attention”, “Language”, “Abstraction”, “Delayed recall” and “Orientation”. “Visuospatial /Executive” consists of three tasks, Trail Making B test (1 point), a clock-drawing task (3 points), a 3-D cube copy (1 point). “Naming” is a naming task with low-familiarity animals (3 points). “Attention” consists of 3 tasks, digits forward and backward (2 points), a vigilance sustained attention task (1 point), a serial subtraction task (3 points). “Language” consists of 2 tasks, a repetition of two complex sentences (2 points), a phonemic verbal fluency task (1 point). “Abstraction” is a two-item verbal abstraction task to explain what each pair of words has in common (2 points). “Memory” and “Delayed recall” is a short-term recall memory recall task (5 points). The short-term memory recall task involves two learning trials of five nouns, face, silk, shrine, lily, and red (“Memory”), and then the delayed recall task is administered after approximately 5 min (“Delayed recall”). “Orientation” is a time and place orientation (6 points).

### 2.5. Sample Size

The sample size was estimated based on the participants’ protein intake. Since sarcopenia and frailty caused by insufficient muscle mass are issues in super-aged societies, an increase in protein intake among older adults is needed. According to previous studies, a 15 g increase in protein intake was estimated to be significant for edentulous Japanese older adults [10,19]. When the significance level, power, and effect size were set to 5%, 80%, and 0.5, respectively, 60 participants were required. Considering a 15% dropout rate during follow-up, the sample size was determined to be 70 participants.

### 2.6. Randomization, Sequence Generation, Allocation Concealment Mechanism, Blinding

Randomization for participant assignment to the dietary intervention or control group was performed using sealed envelopes; the detail of randomization and allocation mechanisms have been described in a previous publication [13]. Clinicians and participants were both blinded to group assignments.

### 2.7. Statistical Method

The Mann–Whitney U test was used to assess between-group comparisons of patient baseline characteristic variables. The Mann–Whitney U test was used to assess between-group comparisons of the MoCA-J total scores at baseline, 3 months, and 6 months. The Friedman test with pairwise comparison was used to test within-group changes in the MoCA-J total scores. Analyses were performed using SPSS 23.0 (IBM Corp., Armonk, NY, USA). A significance level of 0.05 was used.

## 3. Results

### 3.1. Participants

In this study, 79 patients were evaluated for eligibility, and 70 were randomly assigned to the two study arms. Figure 1 shows the study flow diagram and participant assessment. There were 14 dropouts and 56 participants completed all study steps. The baseline characteristics for each group were statistically equivalent, except for age (*p* = 0.045) (Table 1).

### 3.2. Outcome

Table 2 shows the result of the MoCA-J score of each item for the control and intervention group. Table 3 and Table 4 show the results of the MoCA-J total score comparisons between groups and within-group comparisons, respectively. The MoCA-J total score comparison demonstrated no significant differences between the dietary intervention and control groups. However, within-group comparisons revealed significant increases in the MoCA-J total scores between the 3-month and 6-month assessments (*p* = 0.002) and between the baseline and 6-month assessments (*p* = 0.012) in the dietary intervention group. In the control group, no significant increase in the MoCA-J total score was revealed between any of the evaluation periods.

## 4. Discussion

To our knowledge, this study was the first to examine the association between complete denture replacement and MCI improvement and the difference in the effects of complete new denture replacement alone and combined with dietary intervention on MoCA-J score improvement. This study’s results showed that only the intervention group exhibited an increase in MoCA-J scores between 3 and 6 months and between baseline and 6 months, although the scores decreased after 3 months (no significant difference within groups) compares to the baseline. In light of these findings, the study’s null hypothesis was partially rejected.

Banu et al. [20] evaluated the MMSE scores of 10 edentulous patients who underwent complete denture fabrication, implant placement, and implant overdenture fabrication in the edentulous state (without dentures). The results were 17.4, 18.3, and 23.8 points, respectively, indicating a significant improvement in the MMSE scores with complete dentures and implant overdentures. They reported that MMSE scores improved with complete dentures compared to those without dentures. However, the score for those with complete dentures was less than 23, which is considered a low MMSE score, indicating that patients still had cognitive impairment even with complete dentures [20]. The study participants were edentulous patients with severe cognitive impairment and a longer period of edentulousness, which is associated with cognitive impairment. Cerruti-Kopplin et al. [21] reported a significant association between masticatory function and MMSE scores in a cross-sectional study of 117 older adults, showing that wearing well-fitted dentures is associated with a higher cognitive function via higher masticatory function [21]. Furthermore, Kim et al. [22] proposed four mechanisms for cognitive decline mediated by chewing: chewing stimulates the hippocampus directly, increasing the area and number of neurons; chewing increases cerebral blood flow and enhances cognitive function; chewing increases nutrient intake, which enhances the brain function; and damage by inflammatory cells induces brain cell loss, resulting in impaired brain function.

These studies demonstrated that well-fitting dentures can transmit masticatory sensory stimuli from the masticatory muscles, temporomandibular joints, and mucosa to the hippocampus via the trigeminal nerve, contributing to the maintenance of cognitive function. Improving the fit of dentures by fabricating new dentures or adjusting dentures can improve the masticatory function [23]. Thus, for this study, it is suggested that the MoCA-J scores increased in both groups because of the improved fit of the new dentures compared to the old dentures.

Improving the masticatory function of denture wearers alone does not improve food and nutrient intake and that dietary intervention is necessary [10]. Therefore, the mechanism of denture fit improvement by denture adjustment or new dentures for improving masticatory function and cognitive function is not a result of the improvement of cognitive function due to enhancing food and nutrient intake because of the change in eating habit changes but a result of masticatory function improving cognitive function by increasing sensory input to the perioral tissues, such as the mucosa and muscle, owing to increased masticatory stimulation. This study results showed a significant increase in the MoCA-J scores before and after the provision of new complete dentures in only the intervention group, indicating new dentures combined with dietary intervention may have affected the increase in MoCA-J scores. Only the dietary intervention group showed a significant increase in MoCA-J scores between baseline and 6 months and between 3 and 6 months. Therefore, dietary intervention may have been beneficial. It was reported that in the dietary intervention group, the intake of vegetables, fish with bones, and chicken improved significantly in the previous study related to this study [14], meaning that the intake of vegetables, fruits, and meat, which have been considered difficult to chew with complete dentures, increased significantly. Fanny et al. [24] stated that the masticatory muscles’ vascular diameters are rhythmically expanded and restricted during masticatory movements, especially when crushing hard foods. Ono et al. [25] showed that gum chewing activity increases the blood flow in the brain using transcranial Doppler ultrasonography. Onozuka et al. [26] examined the brain activity during gum chewing using functional magnetic resonance imaging and reported a local increase in blood flow because of chewing. These results demonstrate that vigorous chewing may have a beneficial effect on the cognitive function and that the dietary intervention group may have increased sensory stimulation to the hippocampus via the trigeminal nerve and cerebral blood flow more than the control group because of the increased intake of hard foods.

Cognitive screening tests are useful for detecting cognitive decline, and the most commonly used is the MMSE. The MMSE is a 30-point cognitive screening test, with a score of 27 or below defined as suspected MCI and a score of 23 or below as suspected dementia [2]. However, it takes 10 to 15 min to complete and has low sensitivity for MCI. On the other hand, in the MoCA-J, which is a 30-point test, a score of 25 or below defined as MCI and is more sensitive to MCI than the MMSE because it assesses the impairment in cognitive domains, including executive function, attention and concentration, visuospatial abilities, and memory. Trzepacz et al. [27] reported that a score of 25 on the MoCA is equivalent to 29 on the MMSE, and a score of 26–30 on the MoCA is equivalent to 30 on the MMSE [28]. Thus, the MoCA may be potentially more sensitive than the MMSE in detecting cognitive impairment. The MoCA is reported as superior to MMSE for MCI screening in patients with diabetes mellitus and heart failure [29,30].

This study had several limitations. First, only the MoCA-J was used to assess cognitive function. However, it was impossible to determine whether participants had MCI because other assessment tools were not used, and a specialist made no formal diagnosis. Therefore, it is impossible to conclude from the results of this study alone that cognitive function or mild dementia improved because MoCA-J scores increased. Second, there are several risk factors for cognitive impairment, including non-modifiable factors and modifiable factors [8]. Therefore, it is necessary to include these variables as confounding factors in the analysis. Herein, no adjustment was made for all these variables in the analyses. Third, although the participants were assessed as having no cognitive problems, 18 participants in the intervention group and 21 participants in the control group scored < 26 on the total MoCA-J, determined to have MCI at baseline and at 6 months after the intervention; and 14 participants in the intervention group and 17 participants in the control group scored < 26 on the total MoCA-J. It is unclear whether the participants had MCI. Finally, the mean age of the intervention group was significantly lower than that of the control group. It was considered to cause the significant difference in the results for between-group comparisons because age is a risk factor for cognitive impairment [31]. Moreover, it was reported that cognitive function declines significantly after the age of 75 [32]. However, there was no significant difference in the MoCA-J total score between the two groups at baseline, 3 months, and 6 months. Therefore, it is unclear to what extent the 4-year difference between the mean ages of 74 and 78 years in the intervention and control groups influenced the results.

This study should be considered a preliminary study, and a large-scale randomized controlled trial should be conducted with an increased sample size to be able to include all relevant confounding factors in the analyses; expert opinions on cognitive function could also be included in the evaluation.

## 5. Conclusions

This study results showed only the intervention group exhibited an increase in MoCA-J scores between 3 and 6 months and between baseline and 6 months, although the scores decreased after 3 months. The study’s null hypothesis was partially rejected. Considering the limited conditions of this study, it can be suggested that complete denture replacement combined with simple dietary intervention might help improve cognitive screening test results indicated by MoCA-J scores in edentulous older adults.

## Figures and Tables

**Figure 1 jcm-12-04709-f001:**
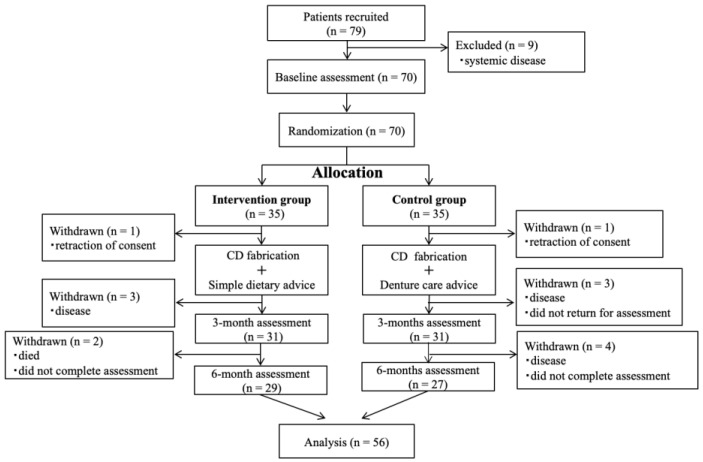
Flow diagram of the randomized controlled trial and participants.

**Table 1 jcm-12-04709-t001:** Characteristics of patients.

	Group		Total	*p* Value
	Control (n = 27)	Intervention (n = 29)		
Age (years) ^a^	79.1 (6.6)	74.7 (8.0)	76.8 (7.6)	0.038 ^c^*
Gender ^b^				0.789 ^d^
Male	14	14	28	
Female	13	15	28	
Body height (cm) ^a^	157.2 (7.8)	157.2 (9.7)	157.2 (8.6)	0.566 ^c^
Body weight (kg) ^a^	55.3 (10.9)	56.3 (11.8)	55.9 (11.3)	0.718 ^c^

^a^ Data were presented as mean (standard deviation). ^b^ Data were presented as number of participants. ^c^ Statistical analysis using the Mann–Whitney test. ^d^ Statistical analysis using chi-square test. * *p*-value for between-group changes.

**Table 2 jcm-12-04709-t002:** MoCA-J score of each item.

	Control Group			Intervention Group		
Item	Baseline	3-Month	6-Month	Baseline	3-Month	6-Month
Visuospatial/Executive (5 points)	4.0 (1.3)	3.7 (1.4)	4.1 (1.1)	4.2 (1.0)	4.0 (1.3)	4.3 (1.0)
Naming (3 points)	2.8 (0.4)	2.8 (0.5)	2.9 (0.4)	2.8 (0.7)	2.7 (0.7)	2.9 (0.4)
Attention (6 points)	4.8 (1.0)	5.0 (1.0)	4.9 (1.1)	5.2 (1.2)	5.0 (1.3)	5.1 (1.1)
Language (3 points)	1.6 (0.8)	1.4 (0.9)	1.6 (0.9)	1.7 (0.9)	1.6 (0.9)	1.8 (1.0)
Abstraction (2 points)	1.7 (0.6)	1.9 (0.4)	1.8 (0.4)	1.8 (0.5)	1.9 (0.4)	1.9 (0.3)
Memory/Delayed recall (5 points)	2.0 (1.6)	2.5 (1.8)	2.6 (1.9)	2.3 (1.8)	2.3 (1.7)	3.3 (0.3)
Orientation (6 points)	5.6 (0.8)	5.5 (0.8)	5.6 (0.8)	5.9 (0.4)	5.7 (0.5)	5.9 (0.4)

Data are presented as mean (standard deviation).

**Table 3 jcm-12-04709-t003:** Comparisons of MoCA-J total score between the control group (n = 27) and the intervention group (n = 29) before and after treatment.

	Baseline			3 Months			6 Months		
	Mean		*p*-Value	Mean		*p*-Value	Mean		*p*-Value
	Control	Intervention		Control	Intervention		Control	Intervention	
MocA-J total score (points)	22.3 (4.0)	23.8 (4.2)	0.089	22.7 (4.2)	23.2 (4.0)	0.575	23.4 (4.0)	25.1 (3.1)	0.105

Data are presented as mean (standard deviation). Statistical analysis using the Mann–Whitney test.

**Table 4 jcm-12-04709-t004:** Comparisons of MoCA-J total score within-group changes of the control group (n = 27) and the intervention group (n = 29) before and after treatment.

	Time Point	Control		Intervention	
		Differences in the Mean Value	*p* Value	Differences in the Mean Value	*p* Value
MocA-J total score (points)	Baseline-3 months	−0.4	0.110	0.6	1.000
	Baseline-6 months	−1.0		−1.3	0.012 *
	3 months–6 months	−0.7		−1.9	0.002 *

Data are presented as the difference in mean value. A negative value for within-group changes denotes an increase in the variable between the time points. Statistical analysis using Friedman test with pairwise comparison. * *p*-value for within-group changes.

## Data Availability

Not applicable.

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
