# Peer review of "Effect of New Complete Dentures and Simple Dietary Advice on Cognitive Screening Test among Edentulous Older Adults: A Randomized Controlled Trial"

_jcm, 2023, doi:10.3390/jcm12144709_

Round 1

Reviewer 1 Report

This study aims at evaluation of the effect of new complete dentures provision alone and with dietary intervention on cognitive functions. They designed a randomized controlled trial, including  70 healthy older adults who required new complete dentures. This study is quite interesting.

I suggest the authors offer more details of the research gap. Also, I would like to know whether there is the possibility that the gender, and age may also affect the observation results. 

Minor reversion 

Author Response

Reply to reviewer #1

As indicated in the responses that follow, we have taken considered all comments and suggestions in the revised version of our paper.

I suggest the authors offer more details of the research gap. Also, I would like to know whether there is the possibility that the gender, and age may also affect the observation results.

Answer:

Thank you for your suggestion. I have added a description of the research gap in the fourth paragraph of the Introduction. Moreover, there was a significant difference in the mean age between the two groups, while there was no significant difference in gender between the two groups. Since there was no significant difference in gender between the two groups in this study, it is not possible to state whether gender affects the MoCA-J total score. However, it was reported that gender is an influential factor in cognitive function (Reference #8).
On the other hand, the mean age of the control group was 78 years, while the mean age of the intervention group was 74 years. It was reported that age is an influential factor in cognitive function (Reference #8 and 31) Moreover, it was reported that cognitive function declines significantly after the age of 75 (Reference #32), Therefore, it was considered that the total score of the MoCA-J of the control group might be lower than that of the intervention group in this study. However, there was no significant difference in the MoCA-J total score between the two groups at baseline, 3 months, and 6 months. Therefore, we cannot state that age affects the MoCA-J total score. Regarding the influence of age on the results, I have added this discussion in the sixth paragraph of the Discussion.

Reviewer 2 Report

Dear Authors

I appreciated the article for novelty and social impact. It Is clinically useful tò know some tips to improve lifes of patients. 

I suggest to insert many more details regarding Montreal Cognitive Assessment in order to help the Reader to understand the way you influenced the cases.

I suggest to remove the useless self-citations. Please they are not professional and useless for this topic.

Best regards 

Author Response

Reply to reviewer #2

As indicated in the responses that follow, we have taken considered all comments and suggestions in the revised version of our paper.

As indicated in the responses that follow, we have taken considered all comments and suggestions in the revised version of our paper.

I suggest to insert many more details regarding Montreal Cognitive Assessment in order to help the Reader to understand the way you influenced the cases.

Answer:

I have added more information on the MoCA-J in the Outcome section of the Material and Methods.

I suggest to remove the useless self-citations. Please they are not professional and useless for this topic.

Answer:

The reference (Komagamine, Y.; Kanazawa, M.; Iwaki, M.; Jo, A.; Suzuki, H.; Amagai, N.; Minakuchi, S. Combined effect of new complete dentures and simple dietary advice on nutritional status in edentulous patients: Study protocol for a randomized controlled trial. Trials 2016, 17, 539. doi:10.1186/s13063-016-1664-y) has been removed.

Reviewer 3 Report

Comments for the Authors:

            Dear Authors,

Your article entitled” Effect of New Complete Dentures and Simple Dietary Advice on Cognitive Screening Test Among Edentulous Older Adults: A Randomized Controlled Trial” bring useful information on the topic and it is also well organized.

I hope that my recommendations will help you to improve the quality of your work. 

In the Materials and Methods section:

2.3. Intervention

- Lines 94-97” The dietary intervention group received new complete dentures and 20 minutes of brief dietary instruction in ...... Dietary Balance Guide for the Elderly published by the Ministry of Agriculture, Forestry and Fisheries [17]”.

o    Please describe in detail the dietary instruction

o    Please verify the cited reference

In the Results section:

-Lines 147-148” Therefore, the null hypothesis was partially rejected.” Please move this phrase to the Discussion section, after paragraph between lines 195-199.

E.g.

” This study results showed a significant increase in the MoCA-J scores before and after the provision of new complete dentures in both groups, indicating new dentures may have affected the increase in MoCA-J scores. However, both groups had an increase in MoCA-J scores after 6 months, while only the dietary intervention group showed a significant increase in MoCA-J scores between 3 and 6 months, although the scores decreased after 3 months (no significant difference within groups) compares to the baseline”. In light of these findings, the study’s null was partially rejected.

- Lines 248- 251” Considering the limited conditions of this study, it can be suggested that complete denture replacement alone or combined with simple dietary intervention might help improve cognitive screening test results indicated by MoCA-J scores in edentulous older adults.” Please move it to the new created Conclusions section 

-Please correct P (uppercase) with p (lowercases) values in all manuscript.

Author Response

Reply to reviewer #3

As indicated in the responses that follow, we have taken considered all comments and suggestions in the revised version of our paper.

In the Materials and Methods section:

2.3. Intervention

- Lines 94-97” The dietary intervention group received new complete dentures and 20 minutes of brief dietary instruction in ...... Dietary Balance Guide for the Elderly published by the Ministry of Agriculture, Forestry and Fisheries [17]”. Please describe in detail the dietary instruction

Answer:

Thank you for your suggestion. The dietary instruction was performed in the following. Brief dietary instruction was orally explained by the dentist using a uniform pamphlet. If the participants did not cook, the person who cooked for the participants received advice by telephone. The pamphlet was a geriatric version of the Japanese food guide published by the Japanese Ministry of Agriculture, Forestry and Fisheries that indicates ‘what’ and ‘how much’ should be eaten in a day. It uses a graphic of a spinning top, a traditional Japanese toy, to illustrate the proportions of each food group that should be consumed daily, similar to the ‘food pyramid’ typically presented to Western audiences. This description has been added in the Intervention section of the Material and Methods.

Please verify the cited reference

Answer:

Thank you for your suggestion. The cited reference has been corrected.

In the Results section:

-Lines 147-148” Therefore, the null hypothesis was partially rejected.” Please move this phrase to the Discussion section, after paragraph between lines 195-199.

E.g.

” This study results showed a significant increase in the MoCA-J scores before and after the provision of new complete dentures in both groups, indicating new dentures may have affected the increase in MoCA-J scores. However, both groups had an increase in MoCA-J scores after 6 months, while only the dietary intervention group showed a significant increase in MoCA-J scores between 3 and 6 months, although the scores decreased after 3 months (no significant difference within groups) compares to the baseline”. In light of these findings, the study’s null was partially rejected.

Answer:

Thank you for your suggestion. I have inserted the description, “This study results showed only the intervention group had an increase in MoCA-J scores between 3 and 6 months and between baseline and 6 months, although the scores decreased after 3 months (no significant difference within groups) compares to the baseline. In light of these findings, the study’s null was partially rejected.” as the last sentence in the first paragraph of the Discussion.

- Lines 248- 251” Considering the limited conditions of this study, it can be suggested that complete denture replacement alone or combined with simple dietary intervention might help improve cognitive screening test results indicated by MoCA-J scores in edentulous older adults.” Please move it to the new created Conclusions section 

Answer:

Thank you for your suggestion. I have created the Conclusion section. I wrote the description, “Considering the limited conditions of this study, it can be suggested that complete denture replacement alone or combined with simple dietary intervention might help improve cognitive screening test results indicated by MoCA-J scores in edentulous older adults.”

Please correct P (uppercase) with p (lowercases) values in all manuscript.

Answer:

Thank you for your suggestion. I have changed the “P value” to “p value” in this manuscript.

Reviewer 4 Report

This research topic is an interesting topic for the aging era.

However, it seems necessary to supplement the notes below.

1. Materials and methods

Is ‘healthy participants’ correct in the Exclusion criteria?

There are limitations in referring to 'healthy participants'.

Age or systemic disease information must be provided.

- Please add detailed description, component for MoCA-J.

- Outcome: ‘A patient was diagnosed ~ ’ Please add a supporting reference for that content.

- Outcome: What is the basis for the evaluation at 3 months and 6 months?

- It is necessary to mention why protein intake is important.

2. Results

- Table 1 Are there any systemic characteristics in the general characteristics?

- Is there a score per item for the MoCA-J?

If a per-item score exists, an item-by-item comment is required.

3. Discussion

- It is necessary to mention the difference between MoCA-J and MMSE.

- It is necessary to check whether the score can be increased through MoCA-J.

 Minor editing of English language required

Author Response

Reply to reviewer #4

As indicated in the responses that follow, we have taken considered all comments and suggestions in the revised version of our paper.

  1. Materials and methods

Is ‘healthy participants’ correct in the Exclusion criteria? There are limitations in referring to 'healthy participants'. Age or systemic disease information must be provided.

Answer:

Thank you for your suggestion. The exclusion criteria were originally written in the Exclusion Criteria section of the Material and Methods. Moreover, we have added the description, “…suffered from severe cardiovascular, metabolic, or other diseases that required specialist dietary restrictions. There were no age restrictions.” in the Exclusion Criteria section of the Material and Methods regarding age and systematic disease.

Please add detailed description, component for MoCA-J.

Answer:

I have added more information on the MoCA-J in the Outcome section of the Material and Methods.

Outcome: ‘A patient was diagnosed ~ ’ Please add a supporting reference for that content

Answer:

I have added reference# 15 regarding the description, ‘A patient was diagnosed ~ .’

Outcome: What is the basis for the evaluation at 3 months and 6 months?

Answer:

Because ethical considerations were required that the dietary advice given to the intervention group should be given to the control group after the study was completed, a longer intervention period than 6 months was not approved. In addition, Because the previous studies that the prosthetic intervention in addition to nutritional instruction (references#11, 12 and the reference (Moynihan PJ, Elfeky A, Ellis JS, Seal CJ, Hyland RM, Thomason JM. Do implant-supported dentures facilitate efficacy of eating more healthily? J Dent. 2012 Oct;40(10):843-50.)) showed significant improvement in nutritional intake 1 ~ 6 months after the intervention, 3 months and 6 months were adopted as evaluation periods.

It is necessary to mention why protein intake is important.

Answer:

In super-aged societies including Japan, the lack of protein intake leads to decrease in the quantity of muscle and may cause sarcopenia and frailty, which is one of the issues that need to be addressed. Therefore, we set protein as the primary outcome in this study to determine whether new denture replacement and simple dietary intervention would contribute to an increase in protein intake for edentulous older adults. The results of the primary outcome are reported in reference #13. I have added the description “Since sarcopenia and frailty caused by insufficient muscle mass are issues in super-aged societies, an increase in protein intake among older adults is needed.” in the Sample size section of the Material and Methods.

  1. Results

Table 1 Are there any systemic characteristics in the general characteristics?

Is there a score per item for the MoCA-J?

If a per-item score exists, an item-by-item comment is required.

Answer:

Thank you for your suggestion.

Although the information on systematic diseases is not shown in Table 1, this study did not include participants with serious cardiovascular or metabolic diseases who were instructed by specialists to follow dietary instructions. Moreover, I have added more information on the MoCA-J in the Outcome section of the Material and Methods. Regarding the results of the items of the MoCA-J, Table 2 shows the mean value and the standard deviation of each item in the Result.

  1. Discussion

It is necessary to mention the difference between MoCA-J and MMSE.

It is necessary to check whether the score can be increased through MoCA-J.

Answer:

Thank you for your suggestion. I have described the difference between MoCA-J and MMSE in the fifth paragraph of the Discussion. Although both MoCA-J and MMSE are screening tests for cognitive decline, MoCA-J is reported to have better sensitivity for MCI than MMSE. Therefore, it is considered possible to evaluate MCI using the MMSE, but the change in the score of the MMSE, which has lower sensitivity to MCI, might be smaller than the change in the score of the MoCA-J.

Reviewer 5 Report

The manuscript entitled "Effect of  New Complete Dentures and Simple Dietary Advice  2 on Cognitive Screening Test Among Edentulous Older Adults: A Randomized Controlled Trial" represents a study already published in 2017 for which additional results (the outcomes at 6 months) are presented.

While the topic of the study is of interest, the manuscript must be improved:

- the abstract should be written according to the journal's indications

- several parts of this manuscript are found written exactly as the previously articles published by the authors in 2017 and 2018.

- In the Introduction section, both previous publications (13, 14) should be described properly, along with their outcomes. Also, the null hypothesis presented must be found in the Conclusion sections for whether is it or not confirmed.

- While it is understandable to cite previous publications in the Methods section, any reader of this manuscript would want to understand completely the Methodology section by reading it. Thus, my suggestion is to add all the information required in order to understand all the activities performed, including the dietary advice,  and the MoCA-J test.

- The Results and Discussion section should emphasize more what are the extra pieces of information provided within this manuscript compared to the previous ones.

- The conclusion section is missing, and it is not clear if the null hypothesis is or is not confirmed

Author Response

Reply to reviewer #5

As indicated in the responses that follow, we have taken considered all comments and suggestions in the revised version of our paper.

The abstract should be written according to the journal's indications

Answer:

Thank you for your suggestion. I have revised the abstract.

Several parts of this manuscript are found written exactly as the previously articles published by the authors in 2017 and 2018.

Answer:

Thank you for your suggestion. I revised the parts in the Material and Methods.

In the Introduction section, both previous publications (13, 14) should be described properly, along with their outcomes. Also, the null hypothesis presented must be found in the Conclusion sections for whether is it or not confirmed.

Answer:

The results of reference #13 and #14 have been added to the fourth paragraph of the Introduction. In addition, the results of the null hypothesis have been added to the first paragraph of the Discussion and the Conclusion.

While it is understandable to cite previous publications in the Methods section, any reader of this manuscript would want to understand completely the Methodology section by reading it. Thus, my suggestion is to add all the information required in order to understand all the activities performed, including the dietary advice, and the MoCA-J test.

Answer:

Thank you for your suggestion. More information including the dietary advice, and the MoCA-J test have been added in the Material and Methods.

The Results and Discussion section should emphasize more what are the extra pieces of information provided within this manuscript compared to the previous ones.

Answer:

Thank you for your suggestion. For the Results, Table 1 has been revised to include only information on the two groups in the present study. Tables 2 to 4 are for outcomes used only in the present study.
In the Discussion section, the results of a study in a related study to the present study are described, which reported an increase in the intake of vegetables, chicken, and fish with bones in food intake three months after the intervention (new denture replacement and simple dietary advice) in the intervention group in the present study, but this report is not the result of the present study. Therefore, to indicate that this report is not the result of this study, we have included the underlined sentence in the following; “It was reported that in the dietary intervention group, the intake of vegetables, fish with bones, and chicken improved significantly in the previous study related to this study, meaning that the intake of vegetables, fruits, and meat, which have been considered difficult to chew with complete dentures, increased significantly.”

The conclusion section is missing, and it is not clear if the null hypothesis is or is not confirmed.

Answer:

Thank you for your suggestion. I have created the Conclusion and described the results of the null hypothesis.

Round 2

Reviewer 1 Report

Agree for publication 

no extra English polish needed

Reviewer 4 Report

The manuscript has been improved. 

Reviewer 5 Report

The manuscript has been improved